# Large-Scale Wind Turbine's Load Characteristics Excited by the Wind and Grid in Complex Terrain: A Review

**Wei Li [1,2], Shinai Xu [3,\*], Baiyun Qian [2], Xiaoxia Gao [3,\*], Xiaoxun Zhu [3], Zeqi Shi [2], Wei Liu [2] and Qiaoliang Hu [2]**

1   Xinjiang Xinneng Group, Urumqi Electric Power Construction and Commissioning Institute, Urumqi 830011, China
2   State Grid Xinjiang Company Limited, Electric Power Research Institute, Urumqi 830013, China
3   Department of Power Engineering, North China Electric Power University (Baoding), Baoding 071003, China
\*   Correspondence: nicenai6666@hotmail.com (S.X.); okspringgao@hotmail.com (X.G.)

**Abstract:** With the development of wind resources under flat terrain, wind farms in extreme wind conditions are developed, and the size of the WT's rigid-flexible coupling components increases. Therefore, accurately understanding the load characteristics and transmission mechanism of each component plays an important scientific role in improving the reliability of WT (WT) design and operation. Through the collation and analysis of the literature, this review summarizes the research results of large-scale WT load under source–grid coupling. According to the classification of sources, the variation characteristics of different loads are analyzed, and different research methods for different loads are summarized. In addition, the relative merits of the existing improvement schemes are analyzed, and the existing problems are pointed out. Finally, a new research idea of 'comprehensively considering the coupling effects of source and network factors, revealing WT load characteristics and transmission mechanism' is summarized. This paper provides important implications for the safety design and reliable operation research of large WTs with complex terrain.

**Keywords:** wind WT; load characteristics; complex wind conditions; power grid; multi-factor coupling; complex terrain

## 1. Introduction

### 1.1. Research Background

Renewable energy continued to grow strongly and, of which, wind recorded its biggest annual increase ever, with 93.6 GW of new wind capacity installed worldwide [1]. 837 GW of WT capacity was installed worldwide, and the world wind capacity was 1,532,000 GWh in the year, with global installed wind capacity almost quadrupling from a decade ago [2,3]. With the falling price of wind power generation equipment, the large scale of WTs is beneficial to the rise in efficiency of wind energy conversion [4]. Flat terrain with abundant wind resources is an ideal site for wind farm exploration [5], which has relatively simple flow field distribution and stable wind speed and, therefore, the power generation and load of WTs are stable and balanced. However, the ideal area is limited [6]. With the increasing demand for clean energy, the development of wind resources and wind farm exploitation gradually turns to offshore and complex terrain [7]. Most complex terrains, such as mountains and hills, are also rich in wind resources. However, in fact, the wind profiles of these terrains are more complex, which results in complicated and changeable load characteristics of WTs and, meanwhile, the increased size of the WT exacerbates the dynamic load characteristics with the rigid-flexible coupling. Some WTs in complex terrain are faced with extreme wind conditions, resulting in large differences in power output and load characters compared to WTs on flat terrain. Simultaneously, due to the increase in tower height, rotor diameter, and the expansion of sweep area, the load on WTs is also larger and more complex, which will directly affect the operating life and the smoothness of the power output [8].

With the enlarged capacity of the large wind farms connected to the grid side, WTs in wind farms also suffering from the effects come from the grid side. For example, the voltage drop of the power grid causes the electromagnetic torque at the generator to fluctuate violently, resulting in dynamic load. The dynamic load is transmitted forward through the transmission system and then affects some positions of WT, such as the spindle and blades [8,9], resulting in additional torque or stress on the transmission system. The simulation result of Marcus et al. shows this phenomenon—that the generator torque shows higher dynamic behavior, and the load on the gearbox increases [8].

For a WT operating on a large scale wind farm in complex terrain, the working conditions of WTs are complicated, which is shown in Figure 1.

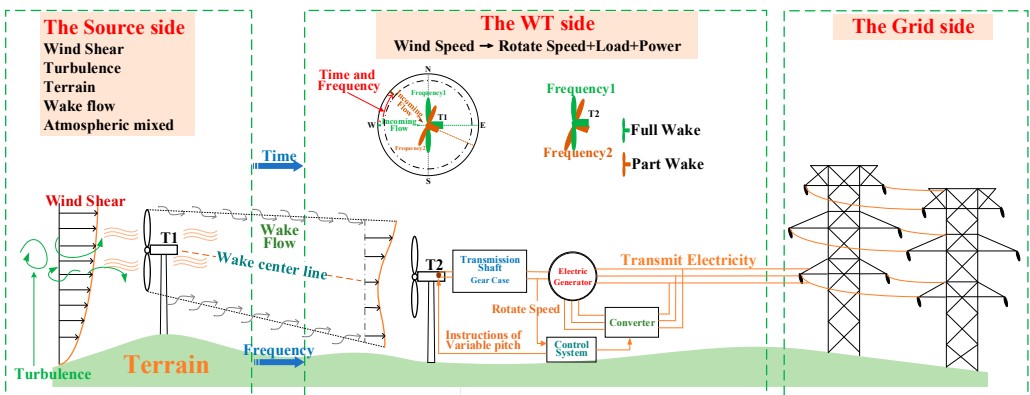

**Figure 1.** Working conditions for WT on a large-scale wind farm under complex terrain.

As indicated in Figure 1, a wind power generation system is complex, including the wind source, the neighboring WTs, and the grid side [10]. In the power transfer chain of 'the wind source—rotor of WT—drive system—generator—grid side', the load of the power transfer chain is complicated, which need to be evaluated accurately. In the non-uniform flow field, the aerodynamic torque generated from the interaction of the rotor and wind is transmitted backwards through the hub spindle [11], while the electromagnetic torque of the generator affected by the transmission system would transmit forward through the transmission system caused by the impact of the grid side. The coupling effects of the double variable excitation from both the wind source and gird increases the complexity of the load characteristics of the WT and the degree of non-linearity in the transmission process.

Accurate evaluation of wind resource distribution and WT load characteristics in complex terrain is one of the key technologies for developing wind farms in complex terrain, which is highly related to the power efficiency and WTs' reliability. Research with different study objects and parameters related to the WT's load was conducted.

*1.2. Literature Review*

1.2.1. Literature Review of the Source Side

For the wind source side, as shown in Figure 1, the flow field in front of the WT is affected by wind shear [12], turbulence [13], wake flow [14], terrain [5], atmosphere mixed [15], etc. Wind shear affects the WT's power loss, the wake variation, as well as the load and service life. Kretschmer et al. [12] noted that the fatigue load on the bending of root blades was affected by wind shear during stable wind conditions, but it was more affected by turbulence during unstable or neutral wind conditions. Turbulence intensity is the main factor influencing the fatigue load of WTs. Higher turbulence intensity and greater turbulence fluctuation will lead to higher fatigue load [13]. In addition, the higher the height, the greater the wind speed. The size and single WT rated capacity of horizontal axis WT also increase [15].

The increase in WT rotor diameter and hub height will also increase the influence of flow structure in boundary layer [16], which has higher turbulence intensity. Turbulence

and the random wind load are also affected by the wake of other WTs [17]. As indicated in Figure 2, the WT's wake can be separated in the near wake region and the far wake region [17]. The near wake region is characterized by double-Gaussian velocity deficit and strong shear [18], while the far wake region is characterized by Gaussian velocity deficit and Gaussian-shaped shear [19]. With the continuous mixing of high velocity atmosphere outside the wake region, the wake velocity increases with the flow direction until the recovery of incoming flow state [20]. The distance between WTs should be selected according to the characteristics of each zone so as to minimize the interaction between WTs. Moreover, the vibration state of the transmission system and gearbox of WT in complex terrain is greatly related to the coupling effect of terrain and wake [21]. The failure rate of aerodynamic components and the transmission system of WT remains high [22]. A probability density function operated in sites with a complex terrain of model WT, which manifested that the WTs need better control systems and robust structures to reduce failure rate [13]. The aforementioned studies present examples for the influences on the WT's load coming from the single parameters of wind resources listed in Figure 1. However, the coupling of two or more factors would lead to more complex WT load characteristics, especially with the temporal and frequency variation of the WT's powering conditioning system been considered. To study the load characteristics and transmission process of WT under dual source–grid variable excitation in complex terrain more accurately, more comprehensive and accurate 'source' side flow field boundary conditions, such as the comprehensive consideration of the above five incoming conditions, are essential [23,24]. Meanwhile, the research of load characteristics excited on the grid side is in progress.

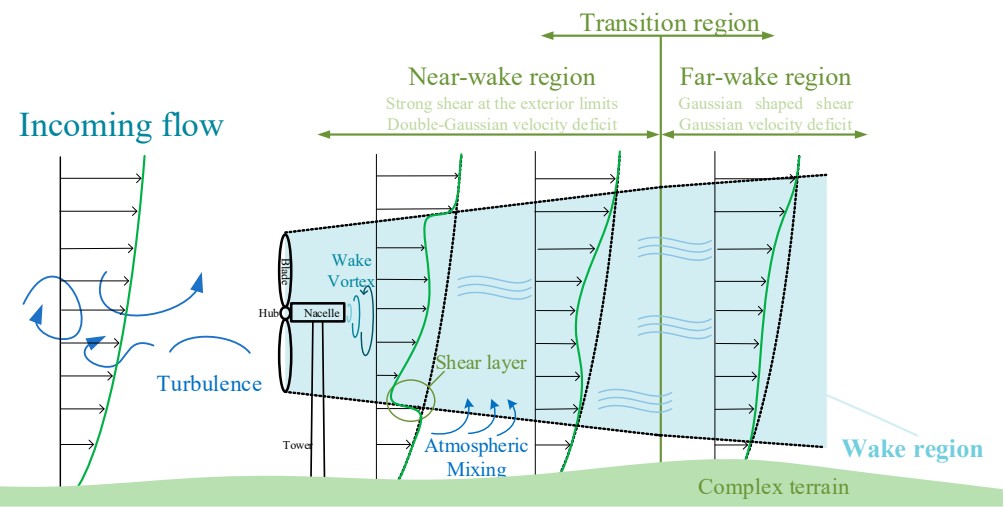

**Figure 2.** Wake structure and partition behind single WT.

### 1.2.2. Literature Review of the Grid Side

For the grid side, WT is a complex mechanical and electrical system integrating machinery, electronics, and electricity. Their operating characteristics are closely related to the characteristics of the power grid. With the increasing installed capacity of clean energy in the power generation system, the electric power generated by WTs with the intermittent, fluctuation, and characteristics of randomness will have an adverse impact on the safe and stable operation and dispatching of power grid [25]. Moreover, whether the wind power can be connected to the power grid smoothly after large-scale development also depends on whether the power system has sufficient peak shaving capacity [26]. The moment carrying capacity of the WT's tower would be dramatically reduced during extreme wind conditions [27,28]. The failure of the grid side not only affects the torsional load of transmission system, but also affects the mechanical dynamic load of other coupling components, such as blades and towers [29]. Voltage drop will directly act on the generator terminal voltage, leading to a significant increase in voltage and current in the rotor

circuit, causing WT speed mutation and the electromagnetic torque of the generator drastic fluctuations [30], thus making impacts on the drive system. During the voltage drop, the output power of the WT will be unbalanced, which will also produce additional torque or stress on the transmission system [31]. With the large-scale development of WTs in complex terrain and the increasing requirements of wind–grid connection, WT load characteristics under dual source–grid variable excitation in complex terrain become more complicated. Therefore, comprehensive summaries and analysis of the large-scale WT's load characteristics excited under multifactor coupling the wind and grid in complex terrain play a critical scientific role in providing adequate information to researchers in related fields, which helps the development of this topic and, furthermore, improves the reliability of WT design and operation and also promotes the efficient and safe application of wind energy.

### 1.3. Innovation Points and Paper Structure

The innovation points of this paper are, therefore, as follows: (a) state-of-art research on large-scale WT's load characteristics are reviewed. (b) The influencing parameters, such as wind and grid, are concluded. (c) The types and transmission process of load are summarized. (d) The normal research methods on this topic were studied. (e) A novel research method for WT's load characters of dual source–grid excitation is extracted.

The structure of this review is as follows: firstly, the load excitation of source side is summarized in Section 2. Closely followed, the load excitation of the grid side is summarized in Section 3. Besides, the multi-factor coupling of WTs is introduced in Section 4. Discussion and conclusions are presented in the last section.

## 2. WT's Load Excitation of the Source Side

### 2.1. Research Status of Source Side Load

There are many studies on the load characteristics of WT under single wind condition on the "source" side, and there are also studies involved in the study of WT load caused by the grid impact on the "grid" side. Unfortunately, the WT load characteristics based on flow field characteristics of complex terrain and its transfer mechanism under the coupling of dual source–grid variable excitation have not been revealed. Simultaneously, the bidirectional transfer law of dual excitation in the energy chain of the rotor–transmission system–generator–grid is ambiguous. Meanwhile, there were many research achievements on load characteristics of WTs considering only source side, the grid side, or complex terrain [32]. Nevertheless, the existing research cannot take a more global approach to comprehensively introduce these problems. Most studies only consider the effect of a single factor, but the WT operating in complex terrain is affected by dual source–grid variable excitation and transference. With the development of wind farms and the increasing flexibility of WTs in complex terrain, the research on load characteristics and transfer mechanism of WTs in heterogeneous flow field needs to be further studied [33]. In particular, rotor torque and tangential force as intermediate parameters of rotor load transfer backwards, and excitation variation characteristics pose a severe challenge to the performance of transmission system. Coupled with the increasing requirements of the wind–grid connection, accurate understanding of load characteristics and the transmission mechanism of each component play a significant part in improving the reliability of WT design and operation. It is requisite to deeply deliberate on the load characteristics of WT under dual source–grid variable excitation in complex terrain.

Therefore, to review the large-scale WT's load characteristics excited by the wind and grid under multifactor coupling in complex terrain plays a scientific part in improving the design and operation of WT reliability. Detailed analysis and conclusions of research on the WT's load characters come from the wind source and are presented below.

The flow field is mostly regarded as quasi-steady [34]. At present, the research on load excitation of large-scale WT includes shear wind, turbulent wind, wind speed disturbance,

yaw wind, yaw shear wind, crosswind, and wake [35–37]. The commonly used research methods are numerical calculation and experiments of wind tunnels and wind farms [38].

As shown in the Figure 3a, the evolution of offshore WT size is moving towards the large-scale. As shown in Figure 3b, the average single capacity of newly installed WTs in China has continued to increase over the years. The large-scale WT is conducive to reducing the cost per kilowatt hour. However, with the increased size of WT such as rotor diameter and tower height, the research based on steady flow is also emerging.

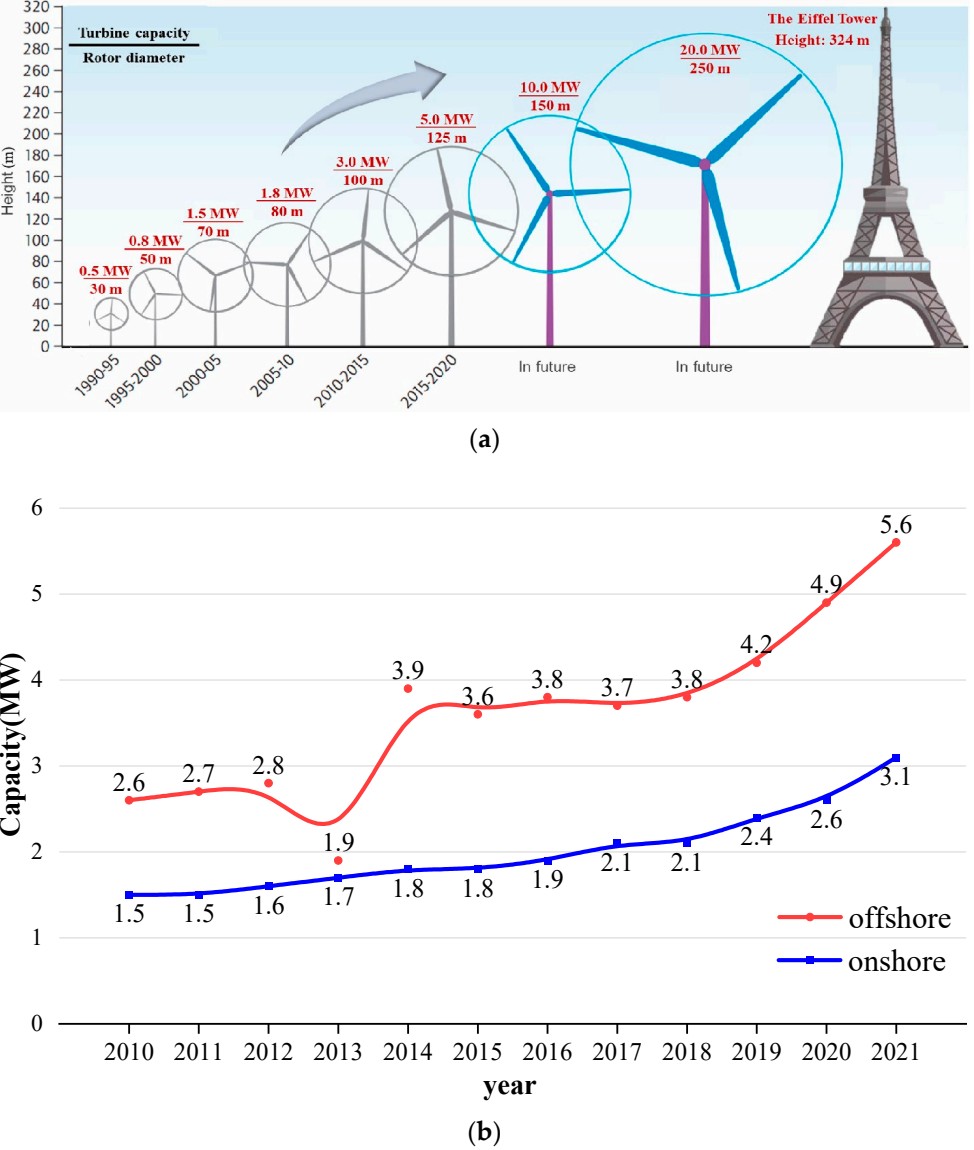

(a)

(b)

**Figure 3.** (**a**) Evolution of offshore wind WT size (modified from IRENA (2021)) [39]. (**b**) The average single capacity of newly installed WTs in China [40].

According to different research positions, load characteristics focus on blade and wind rotor, wind rotor and tower rigid-flexible coupling system, transmission system, as well as the whole WT. Wind farms in extreme wind conditions are being developed [4]. However, extreme wind conditions pose a significant threat to the structural integrity of large WTs. WTs are required to withstand multiple and intense load [41], and the load of standing and random wind may cause colossal damage to internal gear transmission system and the blade [42]. The variation of blade vibration increases with the increase in load WT. The general characteristics of blade vibration change increase with the increase in voltage drop amplitude [43]. All these put forward new requirements for the stable operation and

security of WTs. How to ensure the structural strength reliably and economically is one of the key technical problems to be solved for the development of WTs on a larger scale. Researchers at home and abroad have been exploring and researching constantly in this field. Therefore, they have made plenty of significant studies and valuable achievements. KC et al. [13] established small WT simulation and concluded that WTs operating in the complex environment more possibly have more extreme events caused from larger load fluctuations. In KC experiments, adding a normal turbulence model and ground roughness to flat terrain becomes complex terrain. Furthermore, scholars also add slope as a feature of complex terrain. As the output power of the WT increases, the structure size increases. The structural flexibility of WT rotor blades, support towers, and other elastic components increases, and the dynamic interaction between elastomers may be significant [44,45]. To quantify this interaction, Tien et al. [46] built a test rig to conduct ultimate load tests, which exposed blades to high-intensity cycles to simulate the 20-year life expectancy of WTs. Concurrently, the load of WT can be reflected by deflection, such as the deflection of blades, which reflects the linear distribution of the span of the aerodynamic load [47].

In order to measure the blade deflection accurately, Yongfeng et al. [48] has set up an effective spatial displacement measurement mathematical model on the basis of the geometric transformation method, which can calculate simultaneously the accurate deflection change in three directions. Cazzulani, G et al. [49] monitored the load of WT blade model based on optic fiber sensor. This method can obtain the estimated value of wind load in wind tunnel experiments, thus allowing the optimization of WT control system. Ekry et al. [50] established a WT magnetic levitation system for WT for aeroelastic numerical simulation, which is helpful to reduce WT's load came from vibrations. In order to reduce the operating cost and improve the reliability of WT operation, a load monitoring system consisting of inertial measurement WT is proposed by Wiens et al. [51] to track blade motion at full operation and estimate load. By extracting the ultimate working load to analyze the ultimate strength of WT hub, Zhao et al. [52] researched the sensitivity analysis of the hub's ultimate strength to the mutative load, which provides instructed recommendations for lightweight design and strength safety assessment of the hub. In general, the larger the WT, the more complex the wind conditions and the load characteristics.

Main Sources and Influencing Factors of WT Load

The aerodynamic force acting on the rotor is the main power source of the WT, and it is also the main load source of each component. In addition to aerodynamic load, the other main loads received by WT blades in the operating process of the WT are gravity load and inertial load. The wind power generation system is a complex cycle of strong nonlinear, strong coupling, and time-varying multi-body system factors. The load characteristic is very complex. Excessive load can lead to strong vibration of the WT. In addition, excessive noise [53] will also cause mechanical fatigue damage to components. At the same time, WTs will affect the machine running performance, which will be serious when they act on parts due to fatigue and fracture. On the other hand, the vibration of mechanical parts will also cause the fluctuation of output power and reduce the output power quality of the WTs [54].

WT load is varied in form and complex in source. As in Figure 4, according to the time-varying characteristics of load, it can be divided into five types, namely, stable load, cyclic load, random load, transient load, and resonant excitation load [55]. In the design of WT parts, two loads should be considered: fatigue load and ultimate load [56]. Load can also be classified by source into gravity, inertial, and aerodynamic load, operational load, and other load. The most important are the first three types [57]. These loads have different properties and can be divided into periodic load, static load, steady load, pulse load, transient load, resonant induced load, and random load.

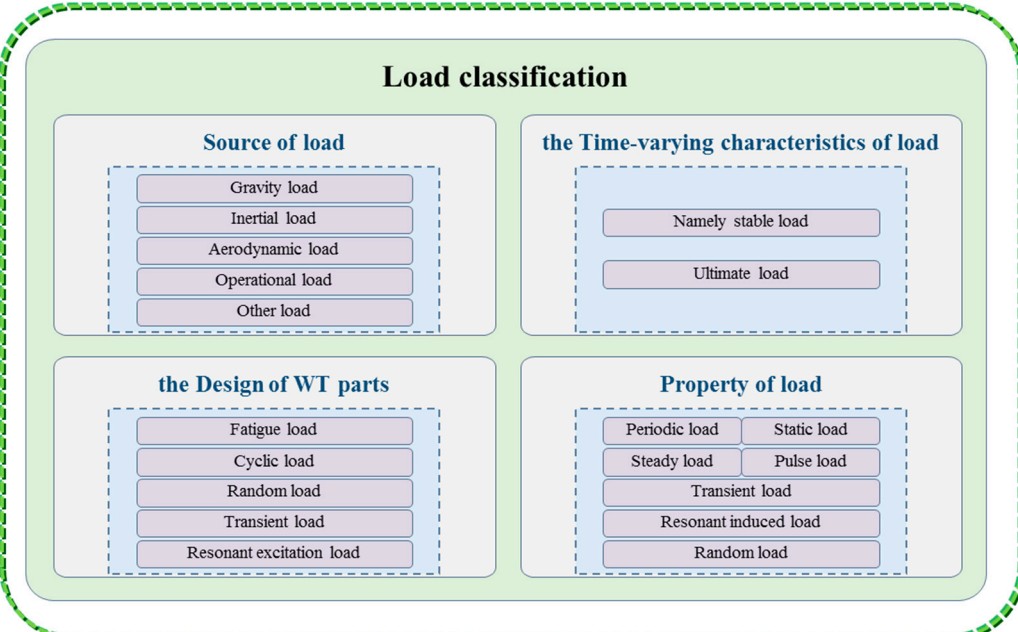

**Figure 4.** Load classification of WT.

Considering the dynamic characteristics of the upper drive system, tower, and coupling, the dynamic load is more complicated. The size and characteristics of the rigid-flexible coupling components of the WT increase, which makes the load more complex. Contrasted with the conventional linear dynamic model, the rigid-flexible coupling effects of WTs is considered in the non-linear coupling dynamic model [58]. Therefore, vibration and deformation are considered, and they are more in line with the engineering reality. Most of the studies on the flow field changes of complex terrain under the coupling effects of the above-mentioned are considered at the level of qualitative analysis. However, the non-linear coupling relationship needs further quantitative research.

Uncertainty of wind–climate parameters affects the WT fatigue load [59]. Excessive wind load may lead to the damage of WT transmission chain components and brake system and generator bearings. In addition, abrupt changes in grid dispatching may lead to WT flying accidents, and the accident may lead to blade fracture or WT tower collapse [60]. When the wind speed is larger than the tolerance limit, the WT will often respond to extreme wind conditions by full feathering or even shutting down. WT influences flow field through the rotor and near wake. However, the load influencing depends on the frequency characteristics of the grid to a certain extent. Wind farms in extreme wind conditions and complex terrain are developed. The load influencing factors and cases are very important in improving the reliability of WT [61].

IEC 61400-1, the WT design specification, as a standard, defines a set of load design conditions suitable for the design and analysis of onshore WTs, which can basically classify and describe the conditions. By subdividing wind speed, abnormal or normal, transport, partial fatigue safety, ultimate strength, global positioning, and other factors can influence the operation result. The standard describes the load design conditions and has great versatility [62].

Studies on the main sources and influencing factors of WT are extensive. In particular, most scholars regard fatigue and aerodynamics as the most complex and related to power output and WT safety, so the current research focuses on the two important parts of fatigue load and aerodynamics load [63,64].

### 2.2. WT Fatigue Load Caused by Source Side Excitation

Cyclic force is a cause of fatigue failure. Each rotation of the WT generates forces that is in complete opposition to the gravity of the low-speed axis and blade, as well as

cyclic load outside the plane of the rotor caused by wind shear, turbulence, yaw error, and shaft tilt. Although the peak value of cyclic load is far less than the safety load, the damage caused by fatigue load cannot be ignored [63]. Severe fatigue failure may result in catastrophic and irreparable damage [65]. Therefore, the design of many WT components, such as gearboxes [66], depends on fatigue load. However, the fatigue analysis of WT cannot be carried out by using simple symmetric cyclic load, but it should consider various effects comprehensively. The unstable conditions, such as sudden changes of incoming flow, correspond to greater turbulent fluctuations and higher turbulence intensities, which also lead to further fatigue load [67].

To calculate the fatigue load of WT [68], finite element analysis software was used to determine the fatigue load and observe the parts of the blade that are easily damaged [69]. Combined with material cycle life and blade load spectrum corresponding to the S–N curve, the Palmgren-Miner linear fatigue accumulation damage theory and Rain flow counting method were applied [70]. Besides, the WT load analysis software GH Bladed was used to select the turbulent model of anisotropy to simulate the fatigue condition, and the blade fatigue load was obtained through dynamic simulation. It is also an efficient method to substitute the fatigue damage surveyed into the probability density evolution equation to calculate the fatigue damage probability density by Rain flow counting method [71]. In addition, Table 1 shows the summary of scholars' research and main contributions to fatigue load in chronological order.

The large capacity WT impeller will rotate $10^8$ times in its 20-year life, ordinarily on complex terrain. Usually, the stress value of $N = 10^8$ cycles is taken as the material fatigue limit. The WT blade is made of fiberglass composite material. The fatigue curve of fiberglass composite material is an approximately straight line with regards to coordinates, and there is no obvious fatigue limit. Aerodynamic force, inertia force, gravity of blades, and the tower will produce coupled vibration, bearing combined load, longitudinal, transverse, and shear strains in different areas of the blade and under different load conditions [72]. Fatigue failure produced by these forces and strains is one of the main forms of WT blade failure, and fatigue failure often occurs at the blade root. It has actual design needs to establish a reliable wind speed model to estimate the fatigue load of blade root and provide a simple and reliable design method. In the process of WT operation, once the blades fail and fracture, the whole generator will be destroyed [72]. Coupled with the large-scale development of wind farms under extreme conditions, the constant change of wind speed and direction leads to the constant change of the mean and amplitude of fatigue load of WTs [73,74], and the fatigue resistance of blades becomes particularly important [75]. Ensure that the structural strength exceeds the stress caused by the applied fatigue load, so that the blade in the life cycle of safe operation [76].

**Table 1.** Information obtained of research on fatigue load.

| Researchers | Year | Object | Analytical Method | Consideration | Main Contribution |
|---|---|---|---|---|---|
| Pehlivan [77] | 2021 | Main load-bearing frame of a 500 kW WT | Conducted stress analysis with finite element method | Fatigue life design, manufacturing, and implementation process | Determined the fatigue and ultimate load of the main load-bearing frame. |
| Jian [78] | 2021 | Blades, hub, and tower of 1.5 MW WT | Rain-flow counting method and data-driven method | Impact of the grid side and the damage equivalent load datasets | Put forward a data-driven method for fatigue load under active power regulation |
| Tian [67] | 2018 | A stationary and rotating model WT | Simulated the change of dynamic wind load under wind tunnel test conditions | In the neutral atmospheric boundary layer | Turbulence intensity dominated the fatigue load of WT. |

**Table 1.** *Cont.*

| Researchers | Year | Object | Analytical Method | Consideration | Main Contribution |
|---|---|---|---|---|---|
| Toft [59] | 2016 | A framework considering the uncertainty of fatigue load is proposed | Structured a probabilistic framework for the reliability level of fatigue load assessment | Speed-up factors, local wind measurements, and distance between the WT and the measuring position | In the structural reliability analyses, uncertainty of wind climate parameters produced fatigue load usually accounts for 10–30% |
| Nejad [79] | 2015 | Multiple transmission system of 5 MW WT | Comparing the fatigue damage | Onshore WT or offshore WT | The main bearing carries axial fatigue load that supports more damage in floating than onshore WT. |
| Vassilopoulos [80] | 2010 | Blades of modern WTs | Fatigue load prediction, random nature of the applied load | Random nature of the applied loading patterns, various material properties | Developed reliable fatigue damage progress models and exploration of fatigue failure by stochastic simulation |
| Kong [81] | 2005 | A 750 KW class horizontal axis WT system | Design and strength verification | Load cases specified at the GL regulations and international specification | Designed a structure of medium scale composite WT blades made by E-glass/epoxy |

### 2.3. WT Aerodynamic e Load Caused by Source Side Excitation

Aerodynamic load, produced by air flow and its interaction with blades and towers, is the source of almost all load and the most important part on WT [82]. The load influenced by the wind conditions of the rotor, the structural and aerodynamic characteristics of the WT, the operating conditions and other factors. In the ideal process of rotation, the plane is subjected to periodic and volatile aerodynamic load. However, affected by turbulence caused by wind shear and wake, the aerodynamic load felt by the rotor varies with time and space [82]. The asymmetry of incoming wind by wind shear makes the rotor subjected to unbalanced aerodynamic load and the interaction between the rotor rotating surface and the wake makes the aerodynamic force of the rotor more complicated. In the interior of complex terrain, the turbulence intensity in the fan zone of wake caused by the wake effect is coupled with the complex atmospheric environment, and the aerodynamic load distribution of WTs in the downstream becomes more complex [14]. WTs affect the flow field through the rotor and near wake. Turbulence will increase the aerodynamic load of local blades and the holistic load of rotor, and the fluctuating transmission of WT load will cause periodic exciting force to other components [83]. The coupling of these factors also makes the overall load of WT more complicated.

Under unsteady aerodynamic load in the actual wind field, because of the blade flexibility and extreme aerodynamic load, in some cases, WT will generate acoustic noise [84], and the blade will even incur large deformation. On the basis of normal vertical wind shear, extreme wind shear can cause changes in wind speed at different points in the rotor plane, usually leading to maximum load on blade section and extreme deformation of blade tip. Aerodynamic models of the full geometry of WTs are needed to optimize its aerodynamic characteristics.

### 2.3.1. Study Models for Aerodynamic Problems Study Methods

The aeroelastic response of a WT is calculated using models that increase its complexity and fidelity [85]. To clarify the effect of flow conditions on WT flow field characteristics and aerodynamic load of WT, the accuracy of the numerical calculation model and method needs to be verified [86]. The mathematical model of complex terrain flow field coupling

atmospheric environment, terrain factors, and WT operation conditions needs to be paid attention to. The WT aerodynamic calculation model can be divided into the following three types: the computational fluid dynamics (CFD) model, the blade element and momentum (BEM) model, and the free vortex wake (FVW) model [87].

Due to the complexity and calculation amount of CFD, analysis can give detailed information about the three-dimensional flow field and aerodynamic performance of the WT [88]. However, because the multi-scale nature of the flow field needs to be considered in the three-dimensional CFD analysis of the WT, it has a large amount of calculation. Although BEM analysis is fast in computation, it needs a lot of operation experience to be modified [89,90]. In addition, the CFD model has preferred flexibility in the blade shape and rotation, as well as in the presence of the tower and nacelle [91].

The blade element momentum method (BEM), which combines the momentum theory and blade element theory and considers the effects of tip loss, stall correction, cascade effect, clearance correction and yaw angle [91], can correctly calculate the aerodynamic performance of the rotor [92], so it has been widely used in WT design and aerodynamic calculation. In addition, the structural parameters of wind shear, yaw, rotor, and WT installation parameters are also considered. Under dynamic conditions, particularly, the dynamic inflow and dynamic stall must be considered [93].

CFD analysis and BEM analysis are used to study aerodynamic problems together. Bangga et al. [94] evaluated the ability of CFD and BEM analysis to predict blade load of 2.3 MW WTs, and the results showed that the CFD and BEM analyses were in acceptable agreement with the experimental data, not only based on average load level, but also in terms of load fluctuation. The difference between the simulated data of CFD and BEM analysis and the measured wind shear and turbulence is less than 10%. Another aerodynamic simulation was accomplished using CFD model based on the finite volume method. Abbaspour et al. [95] indicated that the CFD model can predict the exact geometry with a high precision. Gao et al. [82] analyzed the load of a large capacity WT based on the BEM model, which can help to evaluate the range of turbulence scales that can affect the performance of WT. The aeroelastic response of a 2 MW WT with a rotor diameter of 80 m and interaction phenomena was considered by the use of an accurate unsteady fluid–structure interaction coupling, wherein the BEM model displays, in general, an excellent agreement with CFD model in obtaining the average quantities [85]. It can be seen that the BEM models were used for preliminary estimation, and then the WTs' design was optimized and completed by the detailed simulation of the CFD model [89].

The free vortex wake (FVW) model simulated the attached vorticity of blades in the three-dimensional flow field as concentrated linear vorticity and surface vorticity, and combined with the trailing vortex model, the aerodynamic performance of WTs was analyzed [96]. The three-dimensional flow field information on the blade surface can be calculated, and more importantly, the model does not need too long of a calculation time [64]. To a large extent, the three-dimensional flow field calculation of WTs is simplified, and the computational efficiency of aerodynamic performance of WTs is improved [97]. However, due to the simplification of the CFD/BEM model, the distribution of load on blade surface and the development characteristics of wake are difficult with regards to accurately calculating and characterizing the three-dimensional surface element model of the vortex method. By comparison, the FVW method can obtain satisfactory unsteady aerodynamic load while the amount of calculation is appropriate. Therefore, the FVW method is adopted for contemporary simulations on WT aeroelastic performances. Tang et al. [63] coupled the FVW with the multi-body dynamic method to establish a fast aeroelastic method. Besides, The FVW code wake-induced dynamics simulator is often used to predict the aerodynamic load and wake evolution of offshore floating WTs [98].

The advantage and disadvantage of the three models is compared in Table 2.

**Table 2.** Comparison of the advantages and disadvantages of the CFD, BEM, and FVW models.

| Model | Classification | Point | |
|---|---|---|---|
| CFD | Advantage | 1. | Give detailed information about the three-dimensional flow field and aerodynamic performance of the WT [88]. |
| | | 2. | Better flexibility in the blade shape and rotation or the presence of the tower and nacelle [91]. |
| | Disadvantage | 1. | Having a large amount of calculation. |
| | | 2. | Needing a lot of operation experience to modify it [89,90]. |
| BEM | Advantage | 1. | Calculating the aerodynamic performance accurately of the rotor [92]. |
| | | 2. | Considering the influence of many factors, such as stall correction, tip loss, cascade effect, clearance correction, and yaw angle [99]. |
| | | 3. | The computation is low. |
| | Disadvantage | 1. | Oversimplification leads to an inability to effectively fit the actual situation [89]. |
| | | 2. | Under dynamic conditions, the dynamic inflow and dynamic stall must be considered [93,94]. |
| | | 3. | The blade load cannot be calculated effectively when the wind is not uniform or the pitch angle is not the same in the plane of the rotor. |
| FVW | Advantage | 1. | Could obtain satisfactory unsteady aerodynamic load while the amount of calculation is appropriate. WT aerodynamic performance calculation efficiency is high [100]. |
| | | 2. | Appropriate simplification allows FVW to fit the actual situation [97]. |
| | Disadvantage | 1. | Mainly for the aerodynamic performance of WTs, poor treatment for other aspects [101]. |

The simulation results obtained by the above research methods are usually verified by wind tunnel and wind field experiments [102]. Table 3 shows a summary of key information in chronological order obtained by experimental research by wind tunnel and field. The experimental data are compared with the simulated data to verify the correctness of the proposed theory.

**Table 3.** Experimental research by wind tunnel and wind field.

| Researchers | Year | Experimental Type | Research Purpose | Main Contribution |
|---|---|---|---|---|
| Florian [103] | 2022 | Wind tunnel experiment | To demand well defined closed-loop dynamics to withstand cumulative load over the whole lifetime | Used viewer design and a linear-matrix-inequalities-based control to run a variable-pitch, variable-speed WT in a wind tunnel experiment at repeatable various inflow terms while relying on a derived wind speed estimate |
| Chenzhi Qu [104] | 2022 | Wind field experiment | To determine the direction and value of yaw misalignment | A data-driven calibration method is established and verified in the experiment |
| Fontanes [105] | 2021 | Wind field and laboratory experiment | Examined the electrostatic polarization of electrically isolated WT blades under the effect of fair-weather electric fields | When the blade is immersed in a strong electric field, the charge control system neutralizes the potential gradient at the root of the blade |
| Kan [106] | 2020 | Wind field experiment | To obtain the parameter concerning the potential power output of a WT and a wind farm comprised of a specified number of WTs before installing the WTs | The theoretical distribution of whole farm power is obtained by considering the correlation between the wind speed and WT availability |

**Table 3.** *Cont.*

| Researchers | Year | Experimental Type | Research Purpose | Main Contribution |
|---|---|---|---|---|
| Wei Tian [67] | 2019 | Wind tunnel experiment | Dynamic wind load acting in the atmospheric boundary layer is investigated | More than 90% of the mean and fatigue wind load are caused by the rotating rotor of WT |
| Qing'an Li [107] | 2017 | Wind field experiment | Study the effects of wind shear and turbulence intensity on WT wake characteristics | As the turbulence intensity increases, the maximum velocity deficit in the wake decreases. Meanwhile, as the wind shear index increases, the maximum velocity deficit in the wake increases |
| Arslan Salim [108] | 2017 | Wind tunnel experiment | The wake behind a WT positioned on an escarpment is studied in wind tunnel using particle-image velocimetry | Five different escarpment models were studied, focusing on the sensitivity of WT wake to the geometric details of the terrain |
| Jaeha Ryi [109] | 2014 | Wind tunnel experiment | Development of cost-effective and low noise WT rotor | A prediction method for estimating the noise generated by full-size WT rotors with both a two-dimensional section of the blade and a small-scale rotor is discussed |
| Porté-Agel [110] | 2011 | Wind field experiment | Accurate prediction of atmospheric boundary layer flow and its interactions with WTs and wind farms | Proposed a large-eddy simulation framework and verify its degree of accuracy |
| Migoya [111] | 2007 | Wind field experiment | Derive and verify the relationship among the power output, the wind velocity, and wind characteristics in each WT | The wind characteristics of the measurement situation, the wind speed, the nacelle anemometer, and the power production of each WT are given |

2.3.2. Study of the Effect of Complex Flow Field on the Aerodynamic Load of WT

Flat terrain has few wind resources and is almost completely exploited, so that the development direction of wind resources gradually turns to complex terrain [112]. When the WT works normally in the actual terrain, affected by the terrain, ground roughness, temperature and the wake of front and rear, the atmospheric environment is in a turbulent state, and the WT will be affected by turbulence when working in the turbulent atmospheric environment [113]. With the increase in terrain complexity, wake width variation increases, and velocity deficit exhibition becomes more complex [114]. It is very important to understand the unsteady flow around the horizontal axis of the WT and even the whole flow field. Atmospheric stability affects tower and rotor load [12]. However, the flow around a WT is inherently unsteady [115]. When the rotor bears the unbalanced wind load, such as wind shear, it will increase the overturning moment, which will affect the adaptability of WT to a certain extent. On the basis of normal vertical wind shear, extreme wind shear can cause changes in wind speed at different points in the rotor plane, usually leading to the maximum load on blade section and extreme deformation of blade tip, and wind shear can be divided into positive and negative directions. The extreme operational gust severely affected all the performance parameters of the WT [4].

Wake effect results in fatigue that is larger in horizontal axis WT blades. The fatigue induced by wake has important impacts on the lifespan and efficiency of the whole wind farm [116]. In order to investigate the wind conditions of downstream WT, considerable efforts have been made to optimize the wake model by the relevant personnel. Starting from the initial one-dimensional Jensen wake model, scholars gradually introduced and added parameters, such as wind shear or vorticity [14], and fitted the wind speed curve to super-Gaussian function [116,117] or double Gaussian function [18], and they finally obtained

two-dimensional [14], or even three-dimensiona [33], wake models. The characteristics and expressions of the four typical wake models are shown in Table 4.

**Table 4.** Summary of typical wake models.

| Wake Model | Characteristic | Expression |
|---|---|---|
| Jensen model [118] | Top-hat shape; far wake region | $u^* = u_{hub}\left[1 - (1 - \sqrt{1 - C_T})/(1 + kx/r_0)^2\right]$ |
| 3DJG model [33] | Gaussian shape; far wake region | $\begin{cases} u(x,y,z) = u_{hub}\left[\left(\frac{z+z_{hub}}{z_{hub}}\right)^\alpha - Ce^{-\frac{y^2}{2\delta_y^2}}\right] \\ C = \frac{4ar_0^2}{\sqrt{2\pi}\delta_z r_z}e^{-\frac{z^2}{2\delta_z^2}} + \frac{a\int_{-r_0}^{r_0}\left(\left(\frac{z+z_{hub}}{z_{hub}}\right)^\alpha - 1\right)dz}{r_z} \end{cases}$ |
| 3DEG model [19] | Elliptical Gaussian shape; far wake region | $\begin{cases} u(x,y,z) = u_0\left(\frac{z}{z_0}\right)^\alpha - Ce^{-\left(\frac{y^2}{2\sigma(x)_z^2} + \frac{(z-h_0)^2}{2\sigma(x)_z^2}\right)} - \frac{2a\iint \Delta u\,dA}{\pi r_y r_z} \\ C = u_0\left(1 - \sqrt{1 - \frac{C_T r_0^2}{2\sigma(x)_y\sigma(x)_z}}\right) \\ \sigma_y/D = k_y x/D + \varepsilon_y,\ \sigma_z/D = k_z x/D + \varepsilon_z \end{cases}$ |
| 3DJGF model [119] | Double-Gaussian shape; full wake region | $\begin{cases} u(x,y,z) = u_{hub}\left[\left(\frac{z+z_{hub}}{z_{hub}}\right)^\alpha - C\left(\frac{e^{-\frac{(y+y_{\min})^2}{2\sigma_y^2}} + e^{-\frac{(y-y_{\min})^2}{2\sigma_y^2}}}{2}\right)\right] \\ C = \frac{4ar_0^2}{\sqrt{2\pi}\sigma_z r_z}e^{\frac{y_{\min}^2}{2\sigma_y^2}}\left(\frac{e^{-\frac{(z+z_{\min})^2}{2\sigma_z^2}} + e^{-\frac{(z-z_{\min})^2}{2\sigma_z^2}}}{2}\right) + \\ e^{\frac{y_{\min}^2}{2\sigma_y^2}}\frac{a\int_{-r_0}^{r_0}\left(\left(\frac{z+z_{hub}}{z_{hub}}\right)^\alpha - 1\right)dz}{r_z} \end{cases}$ |

As shown in Figure 5, The engineering wake model is becoming closer and closer to the measured results.

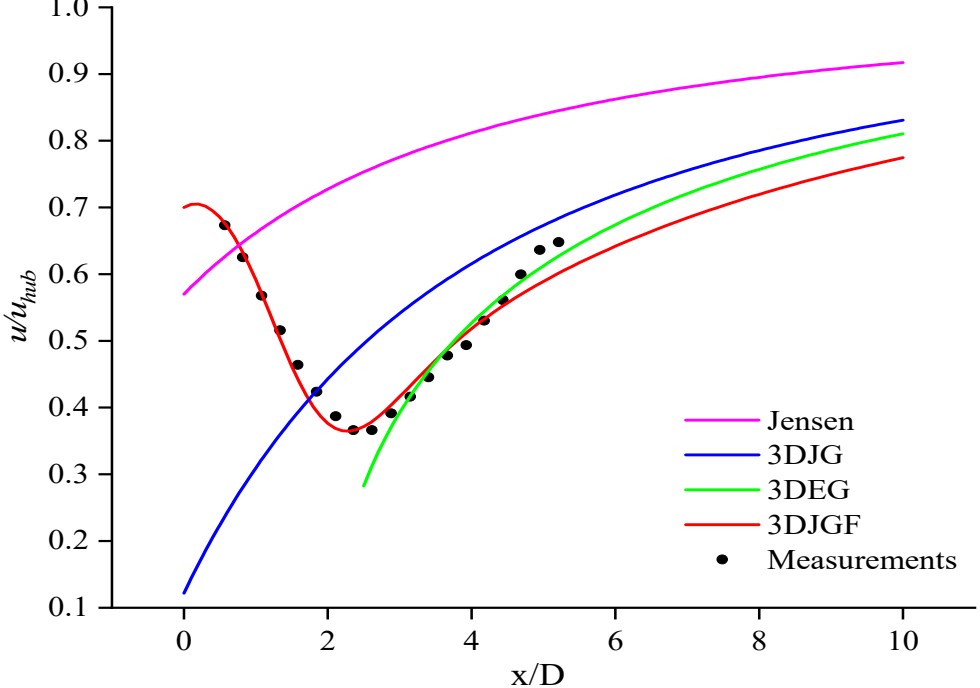

**Figure 5.** Comparison of wake model prediction results and experimental data on wake centerline in the range of x = 0~5D [119].

Results from interactions between turbulence with the highly three-dimensional flow on blades and the inflow conditions, the field is very complex [4,94,120]. WTs are sub-

ject to instantaneously varying load due to wind turbulence [121]. Higher inflow turbulence results in greater power fluctuation [36]. Due to the extreme wind shears, the blades experience asymmetrical surface pressure variations [122]. Flow visualization techniques with tufts and oil film methods are effective for investigating the stall flutter of horizontal axis WTs, and they also offer a better understanding of the development flow [123]. Meanwhile, Qing'an et al. [123] proposed a more exact visualization of flow field and aerodynamic load on horizontal axis WTs in turbulent inflows. Furthermore, extensive scientific studies show that the fatigue load of WT is increased by about 80%, and the power loss caused by the wake to the downstream WT is about 40% [88,116,124]. Greco et al. [125] studied the performance of a large-scale WT in yawed flow and axial flow investigated by a free-wake, unsteady three-dimensional aerodynamic model. Numerical results concerning experiments and other comprehensively used methods show that the panel method rotor aerodynamics were rigorous, thus avoiding time-consuming CFD analysis. Hanssen-bauer et al. [126] developed an accurate and fast tool for simulating the load and performance of WTs in the flow field by using the Mann model, a reduced order model, as well as an inflow mode based on LES data. Macrí et al. [37] measured the load variation on a downstream WT with unsteady aerodynamic load balance. Conti et al. [127] combined measurements of nacelle-mounted LIDAR with wake of wind field reconstruction technique to improve the accuracy of WT load assessment under wake conditions. Most of the above studies are simulation or wind tunnel tests, and the data obtained, such as describing wind flow accurately through these methods, are helpful for the study of wake and load. However, it is not realistic to solve the actual layout of wind farms through these studies. The wind tunnel test is essentially a practical simulation. When optimizing the positions of WTs in complex terrain, all optimization simulations require much time of calculation for the wake low. Coupled with the wake interaction effect [127], those make the design process for applying it to wind farms difficult.

### 2.3.3. Aeroelastic Phenomenon

The complexity of wind load enhances the complexity of aerodynamic characteristics and affects the power output. The increasing size of the rigid-flexible coupling components of WTs have been proposed [128], resulting in severe aeroelastic effects caused by the interaction of elastic deflections, inertial dynamics, and aerodynamic load [129]. Meanwhile, the aeroelastic phenomenon has a significant effect on the structural response and wake distribution of WTs [64]. The induced velocity of blade deflection in the flatwise direction further deviated the apparent wind speed experienced by the flexible blade, resulting in a time lag between the prescribed motion and its aerodynamic load [130].

The performance of a flexible model is reduced due to the aeroelastic effects. Compared to the rigid model, the attack angle of sections of the flexible model was lower. The aeroelastic effects also lad to aeroelastic instability problems, such as flutter and edgewise instability, which can be ruinous to the blades, or even the whole WT, when the size of WT becomes larger and larger [86]. Fluid structure interaction also has critical influence on aerodynamic load. Aerodynamic performances showed more asymmetry under the combined effect of fluid structure and yaw coupling [47].

The main content of aeroelasticity is the unsteady interaction between structure, inertia, and aerodynamic phenomena, which is beneficial to the design of anti-fatigue blades. Stall flutter is an aeroelastic phenomenon that generates unwanted oscillatory load on WT blades. Stall usually occurs at a high angle of attack [95]. Li et al. [131] presents an aeroelastic model and simulated the effects under different stall conditions, and the control algorithm of the stall flutter was investigated.

Horizontal WT mainly relies on blades to capture wind energy and convert it into electricity [75]. One can reduce the cost per WT length of the blade by realizing adaptive load reduction of the blade through bending and torsional coupling control. One can improve the aerodynamic damping in the blade face through a reasonable material arrangement scheme to improve the blade reliability [132]. The design scheme of a flexible

blade with pneumatic accessories can reduce the risk of blade stall and ensure the power generation of the WT.

2.3.4. Optimization of Aerodynamic Characteristics

Wind farms in extreme wind conditions are developed, the size of rigid-flexible coupling components of WTs has increased, and the requirements for wind–grid connection have been improving. Optimization of aerodynamic characteristics plays a vital part in improving the reliability of WT design and operation [129]. For example, blade optimization can improve the power coefficient at the design tip speed ratio [133].

Akram et al. [134] analyzed the aerodynamic characteristics of an asymmetric WT airfoil of NREL S809 by parameter and class shape transformation. The existing method indicates that the optimized airfoil by class shape transformation is predicted, with 9.6% for the lift coefficient and an increment of 11.8%, as well as desirable stability parameters obtained for the design of WT blades, respectively. These characteristics dramatically improve the optimization of aerodynamic characteristics. For variable-speed WTs, Minghui et al. [135] proposed a multi-point aerodynamic design method with an objective function that can approximate the wind energy capture efficiency, aiming at improving closed-loop performance and better harmonizing static aerodynamic performance and the maximum power point tracking dynamics, so as to obtain higher power generation of variable-speed WTs.

Horizontal WT blade icing occurs in extreme wind conditions [136], which will obviously affect the aerodynamic characteristics of WTs [137,138]. Icing transforms airfoil profiles of the blades and exacerbates the aerodynamic performance of WTs, and is consequently results in power production loss [139,140]. It is necessary to predict the icing shape of the airfoil, to analyze the aerodynamic performance of airfoil, and to optimize the design of the airfoil under icing conditions [141]. To study the mechanical behavior of ice thickness, Lagdani et al. [142] suggest a numerical model to simulate 50 mm ice thickness located in the tip of a horizontal axis WT blade. Yirtici et al. [138] proposed an optimized aerodynamic shape of WT blades to minimize power loss due to icing. Xu Zhang et al. [143] developed a WT blade suitable for ice and frost environment through blunt trailing edge optimization design, which reduced the adverse effect of ice on aerodynamic characteristics.

## 3. WT Load of Variable Excitation of the Grid Side

### 3.1. The Effect of the Grid on WT Load

In the actual work, the wind–grid connection needs to control the overall voltage of the wind farm, fully consider the basic conditions, and then adjust the voltage deviation of the whole power grid system according to the actual power, so as to guarantee the reliable operation of the whole voltage system [144]. To deal with the large-scale access of wind power, it is a requirement to ensure that the WT is in normal state under extreme wind conditions and grid dispatching conditions.

The power of WT and the power of grid load both have fluctuation characteristics, but the actual operation experience shows that there is little correlation between the fluctuation trends of the two [145]. The fluctuation of wind power will increase the power adjustment burden of other generators in the power grid and affect the economy and security of other conventional generators. For the load caused by power fluctuation, both sides of source and grid must be considered, as they are the key to maintain the safe stability of the power grid and to realize the optimal allocation of power resources.

With the massive consolidation of wind power, the power grid requests WTs to be equipped with low voltage traversal function. That is, when the voltage of the grid side has a large fluctuation or decreases to a certain value, the WT needs to be guaranteed to operate without leaving the grid, and reactive power can be provided to sustain the restoration of the grid [146]. Analyzing the dynamic performance of crowbar resistance under symmetrical and steady-state grid disturbances, Gebru et al. [147] performed the crowbar resistance conservation on the actual large wind farm to strengthening the low

voltage traversal function. In the process of the low voltage ride through, the blade vibration has obvious impact because the load oscillates. The vibration amplitude of blade flapping direction is more obvious, and the load change of blade flapping direction will cause the axial load change of the whole WT, affecting the safety and reliable operation of the WT. Therefore, the requirement of the power grid for the low voltage traversal characteristics is that the WT must operate under the power grid fault state. This poses a severe challenge to the transmission system and even the whole system. To provide tendency and extent of the occurrence of voltage sag anticipated to trigger WT disconnection and to accurately assesses voltage sag condition at connection points of WT, Shakeri et al. [148] proposed an iteration-based technique to calculate expected sag frequency with high accuracy considering low voltage traversal characteristics for WT.

Severe power failure on the grid by extreme gusts will cause the blade bending moment to reach the ultimate load. The power failure not only affects wind rotor speed and variable blade angle, but also affects the load of the whole machine. As a result of power failure, loss of power, or the quality of electricity, it does not meet the requirements. Thus, the power grid side cannot work normally, so the generator side offs the load. This leads to a rapid increase in the speed of the WT, and then the speed decreases with the increase in the variable blade angle and the decrease in wind energy absorption by the rotor. It can be seen that the moment of power fail has a great influence on the bending moment of blade root, and certain strategies should be adopted in the design or actual operation to ensure the safety of the WT under various possible conditions.

### 3.2. Review of the Grid Side Failure

Voltage drops when the power grid malfunctions, such as in single grounding short circuits, two grounding short circuits, 2-phase short-circuits, or 3-phase short-circuits. This will cause a sudden change of the rotor speed and the drastic fluctuation of the electromagnetic torque of the generator, which will impact the mechanical parts, such as the transmission system of the WT. Voltage drop will directly act on the generator terminal voltage, leading to a significant increase in voltage and current in the rotor circuit, causing the WT speed mutation and the electromagnetic torque of the generator drastic fluctuations and impact on WT drive system. During the voltage drop, the output power of the WT will be unbalanced, which will produce additional torque or stress on the transmission system [31]. Meanwhile, the changes of electromagnetic torque and output power will also act on the grid side in reverse. Under normal circumstances, even if the grid has a minor fault, the WT may be disconnected from the power grid.

With the development of large-scale single WT, the torque of the transmission system increases, and the dynamic load caused by transient faults of the power grid has an increasing influence on the parts of WT. Meanwhile, the vibration problem of WT is gradually more prominent. The WT must be connected to the grid for a period of time under the voltage sag of the grid, and the WT can quickly recover to the normal working state after the grid fault is removed [149]. However, in the transition process of power grid voltage sag and recovery, the electromagnetic torque of the motor will fluctuate greatly, which will inevitably bring about the oscillation of the torque after the fault process and fault removal, and it may further impact the mechanical components, such as gear case, affecting the operation and life of the WT [150]. At the same time, it may influence the stability of generator output power and speed.

### 3.3. Effect of Grid Failure on WT Load

The dynamic characteristics of WT will affect the grid-connected quality of wind power, and the interference and faults of the power grid will also affect the mechanical and electrical components and mechanical components of wind power. When the grid side runs normally, it has little influence on the source-side load, and the variation of the source side load caused by the voltage sag of the grid can be recovered in a very short time. The simulation analysis of Ying Ye et al. [151] expresses that 3-phase short-circuit happens on

the grid side operations, which have less influence on the stabilization of doubly-fed WTs. The changes in grid behavior are mainly reflected in the voltage, the power flow, and the system frequency. The need load of WT is affected by the voltage sag, the variation of the direction, and the value of load flow [152].

When the power grid disturbance or fault leads to the voltage sag of the wind farm connection point, the WT can operate continuously in a certain range of voltage sag [153]. The transient component of the motor flux excited by the voltage sag of the power grid causes the electromagnetic torque of the generator to oscillate and then causes the mechanical dynamic load response of the WT [154]. Due to the large inertia of WT impeller, the impeller speed may not be affected [155]. However, the transient voltage sag rapidly changes the blade tip acceleration, especially in the direction of shimmy, and increases the mechanical fatigue load of blades, spindle components, and tower, while the shear force of spindle and shear force in other directions of tower are not affected. At the same drop depth, the greater the voltage value at the drop time, the greater the impulse current. After the recovery of the transient voltage sag, mechanical parts need a long time to recover to a stable condition because of the limitation of inherent damping [31]. The torque pulsation caused by voltage sag will eventually produce mechanical fatigue load in mechanical components and affect component life. Simulation results show that the rapid recovery of voltage transient is helpful to alleviate the mechanical dynamic load of the WT. The voltage sag of the power grid causes the imbalance between mechanical and electrical power of WTs, which makes the load of WTs oscillate, and in serious cases, the WTs are cut off. Therefore, the WT load-related test should also be carried out when the WT is tested at low voltage.

When a power grid fault occurs, it will inevitably lead to the change of WTs on stator current, rotor current, and speed. In order to calculate the maximum fault electric present expressed of the rotor currents, L Yu et al. [156] introduced a direct method based on space vector to acquire an exact representation of rotor electric currents as the time function during the voltage failure. Extreme operating conditions even cause a loss of connection between the source and grid. The load from sudden or continuous interruption of electrical connection imposed on large offshore WTs may result in increased rotor speeds or emergency shutdown [157]. In reference [155], a fixed-speed WT model was built to conduct simulation research on the variable speed and rotor WT under power grid failure, but there was no study on the actual commonly used WTs. For common WTs, scholars have built a co-simulation model [158] based on FAST and MATLAB/SIMULINK software to simulate the output speed of WT and the load of key parts in the case of power grid failure. The simulation results show that permanent magnet synchronous generator is more effective than the squirrel cage induction generator and the doubly fed induction generator for low wind speed [159].

## 4. The Multi-Factor Coupling Effects on the WT's Load Characteristics

### 4.1. Research Model of the Transmission Chain

The WT drive train is a complex multi-body coupling system, and its dynamics characteristics have important effects on the stability and reliability of WT. In its life cycle, WT is excited by complex wind load of random variation for a long time, which is the main load source of the whole WT. These loads will be transmitted back to the transmission system and even the generator through the main shaft. It means that WT load is transmitted through both sides of the transmission chain under dual source–grid variable excitation. It is known from the theory of classical mechanics that the rotor is subjected to both aerodynamic torque and low speed shaft torque, resulting in the dynamic process of rotor rotation. The dynamic process of generator rotation is produced by the resultant force of load torque and high-speed shaft torque. As shown in Figure 6, the transmission chain of WT is generally composed of blades, hubs, low-speed drive shafts, gearboxes, high-speed drive shafts, and generators.

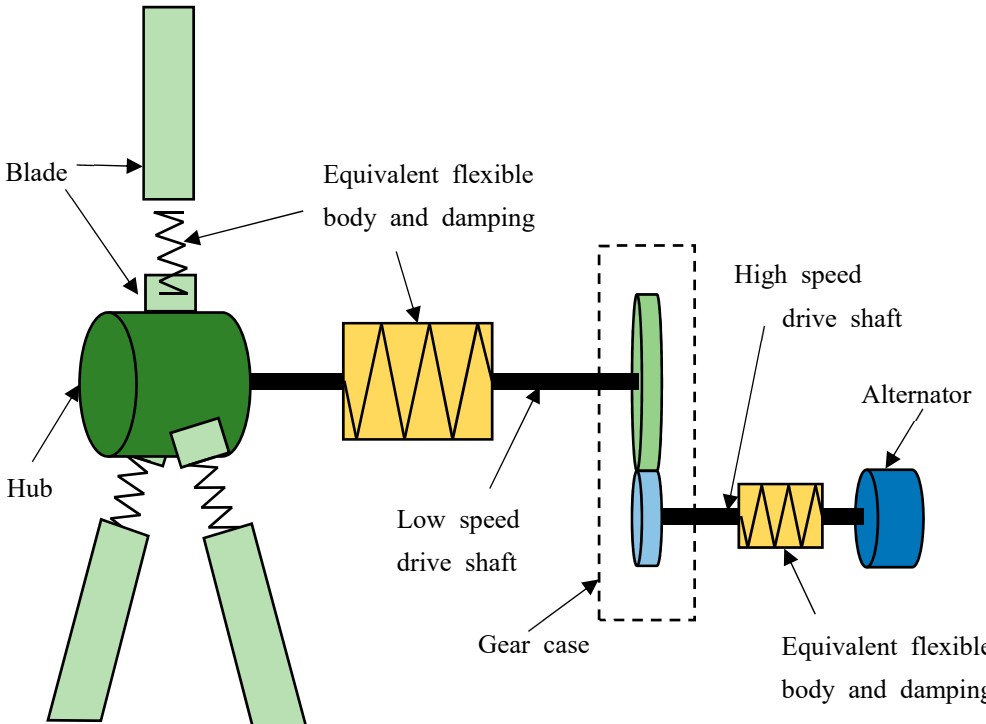

**Figure 6.** Diagram of a typical WT transmission chain.

4.1.1. The Multi-Mass Block Equivalent Model of WT

It is pointed out that all rotating shafting systems can use several mass blocks to describe the physical parameters of the real system, and the more the number of mass blocks and degrees of freedom, the greater the model precision [160]. For the multi-mass block equivalent model of different research contents, the equivalent mass block is often different. Some parts of the transmission chain are equivalent to mass blocks, and the equivalent model of the mass block transmission chain of WTs is obtained [161]. Some models also consider the flexibility of drive chain shafting. Considering the above factors, the mass block model is widely used [162]. Typical research is reviewed and presented in this paper.

The 1-mass block equivalent model takes the whole transmission chain of WT as a rigid structure, which is equivalent to a mass block. In most cases, WT's single-mass model is used for controller design

The method of 2-mass block equivalent model could be used to equate the WT and generator into two mass blocks and then connect the WT and generator through a gear case, as in Figure 7a. The 2-mass block equivalent model also could equate the blade and hub to one mass block and the gearbox and generator rotor to another mass block, as in Figure 7b. The 2-mass block equivalent model can even equate the gearbox and generator as two mass blocks, which are then connected behind the hub [163], as in Figure 7c. In the 2-mass block model, the blades, hub, gearbox, and generator are equivalent in the mass block, and then they are connected with the rest through a low/high-speed shaft. Vijay P. et al. [162] has investigated a small-signal stabilization analysis for the WT system on the basis of the above 2-mass transmission train models and its joint model with thermal power integrated system. Besides, by using the 2-mass model, an algorithm of adaptive backstepping has been obtained to implement the progressive generator with speed tracking [164]. Muyeen et al. showed that the dual-mass axis model was sufficient for transient stabilization analysis of wind power generation system with reasonable accuracy [165].

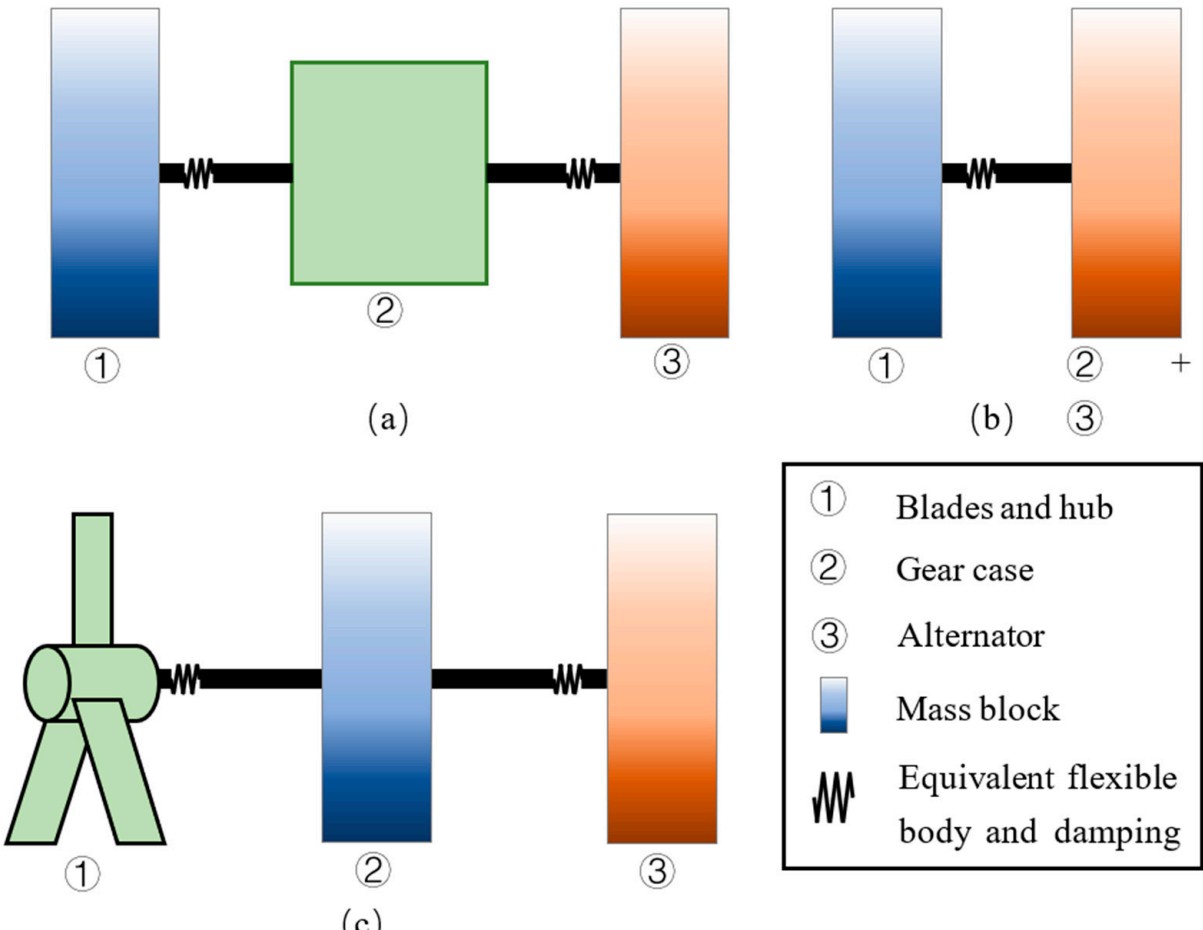

**Figure 7.** The 2-mass block equivalent model. (**a**) Case 1 in The 2-mass block equivalent model; (**b**) Case 2 in The 2-mass block equivalent model; (**c**) Case 3 in The 2-mass block equivalent model.

The commonly used 3-mass block model is used to separate the equivalent planetary gear train from the generator module to form a 3-block model. A mass block represents the WT, a mass block represents the planetary gear train module, and the last mass block is the generator module. Another commonly used 3-mass block model considers blade flexibility and low-speed shaft flexibility, and three mass blocks are used to represent the blade and hub, low-speed shaft, and high-speed shaft, respectively [166]. This model may be more suitable for short-time accurate harmonic evaluation on the grid side [167]. As shown in Figure 8, there is also a 3-mass block equivalent model of 3 mass block transmission chain for WT. Blades and hub, gear case, and alternator on the transmission chain are three mass blocks. Considering more than one discrete mass modeling for the drive train, M. Seixas et al. [160] constructed a model for offshore WT system by simulations with 1, 2, and 3-mass drive train modeling. Meanwhile, based on the 3-degrees-of-freedom (3DOF) mathematical model, Jing et al. [168] established the simulation model of 3DOF rigid-flexible coupling multi-body dynamics.

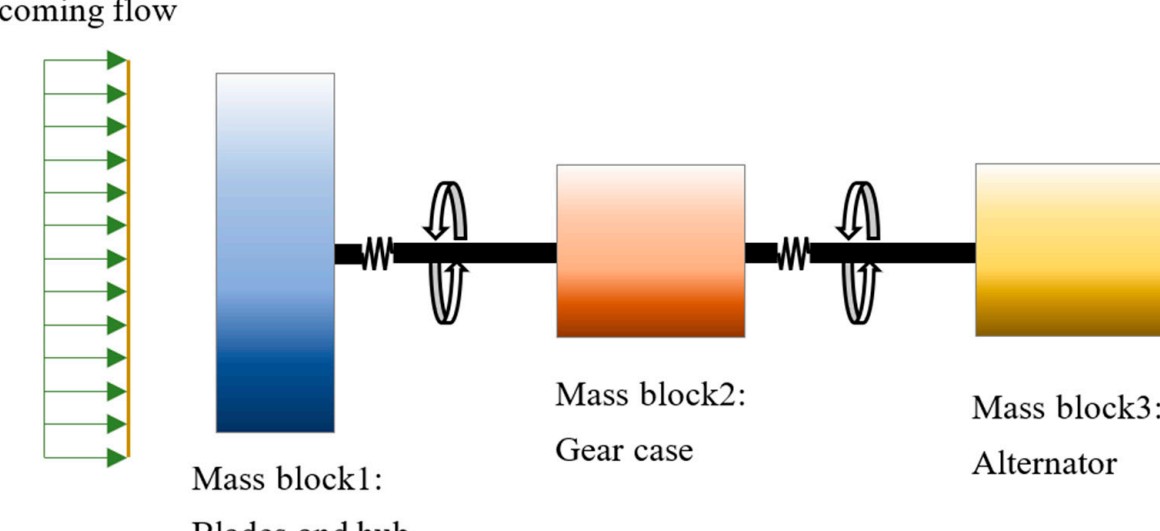

**Figure 8.** Equivalent model of 3-mass block transmission chain for WT.

The mass block model of the drive chain mostly focuses on the single wind condition or the simple wind condition. However, the WT load in the unsteady and heterogeneous flow field in complex terrain is less involved. In particular, the spindle torque and axial thrust of the WT in the complex flow field will change with the rotation cycle and phase angle of the blade and the uneven flow field in front of the rotor. This directly leads to the unstable changes of shaft and gear, gear, and gear meshing in the transmission system of the WT, which affects the operation stability of the transmission system.

### 4.1.2. Flexible Multi-Body (FMB) Model of WT

The FMB model of WT is also an effective method to study the transmission chain. The dynamic characteristics parameters of components and the whole transmission system is obtained by simulation analysis under the WT dynamic of the FMB model. Some components in the transmission chain are made into flexible bodies by the finite element method and by combining with the system by the multi-point constraint method, and the multi flexible body dynamic model of transmission chain is obtained. Meanwhile, the simulation of vibration characteristics was carried out so that the aerodynamic load and the torque were applied to the structure of the rotor. Simultaneously, the deformation of the structure also affected the load. Jiao He et al. [169] established a FMB dynamics model of a floating WT in SIMPACK to control the structural vibration of the floating WT. Through numerical simulation on dynamic response of the FMB model, Cheng et al. [170] ascertained the need for modified structural components by monitoring multiple parameters of WT tower/blade coupling structure under various operating conditions.

The transmission chain of WT contains many components, and the corresponding parts will be flexible. Some of components have complicated structure and various materials, which leads to a mesh quantity of the model that is so large that it is not conducive to solving the calculation. The rigid/flexible body modeling mode and freedom selection of each component are shown in the Table 5, according to GL2010.

**Table 5.** Modeling standards of transmission components.

| Component | Rigid or Flexible Body | Degrees of Freedom |
| --- | --- | --- |
| Blade | Flexible body | 2 |
| Hub | Flexible body | 6 |
| Box of Gear case | Rigid body | 6 |
| Planet carrier of Gear case | Flexible body | 6 |
| Gear bearing of Gear case | Flexible body | 6 |
| Gear of Gear case | Rigid body | 6 |
| Coupling | Rigid body | 6 |
| Generator rotor | Rigid body | 6 |
| Generator stator | Rigid body | 6 |
| Rack | Flexible body | 0 |

*4.2. Control Strategy WT's Load Reduction under Multi-Factor Coupling*

Comprehensively considering the coupling effects of source and grid factors reveals WT load characteristics and transmission mechanisms. In addition, when the actual wind farm is connected to the grid, multi-factor coupling (source, wind rotor, main shaft, transmission system, generator, and grid) will make the load of system and grid connection frequency of the WT more complex. On the one hand, the increase in the proportion of new energy generation will lead to the reduction of the power system inertia [171], thus worsening the frequency stability of the power system [172]. On the other hand, under multi-factor coupling, the WT can compensate the power system inertia decline by inertia control. However, this inertial control often varies the generator torque of the WT according to the system frequency, thus increasing fatigue load on the spindle. In order to decrease the influence of inertia control on fatigue load of the WT, Wang et al. [173] proposed a proportional-integral (PI) mitigation-based control solution, which improves the inertia response capacity of WTs while vastly reducing the fatigue load of the main shaft. Edrah M [174] evaluated the impacts of the inertia controller on the WT structure, and the results show that the implementation of inertia controllers on the full-scale converter WT can improve the inertia response, but will affect the dynamics of its blades, drivetrain, and tower. Fortunately, these effects are controlled are have been shown to be smaller than the effects caused by some grid faults.

The variation of the axial induction factor was controlled by varying the pitch angle and other methods to improve the residual wind energy in the wake, as is a commonly used method. Another method, active yaw control (AYC), has great potential in wind farm power and load optimization [175]. The total power output can be augmented by installing WTs upstream. The active control of WT is more consistent with the actual operation of the wind field [176]. The wake can be offset by the yaw of upstream WT to improve the output power of downstream WTs. Although the yaw of the upstream WT reduces its own power, its wake offset can effectively improve the output power of the downstream WT. Adaramola et al.'s [177] research shows that the yaw operation of the upstream WT can change the distribution area of wake flow, reduce the influence of wake flow on the downstream WT, and significantly improve the product power of the downstream WT. The significant effect brought by this technical means is that the total product power of the 2 WTs can be increased by about 12%.

Speed and pitch angle can be considered in optimal control of wind farm under wind power curtailment. When the rotor speed control can meet the demand of the wind curtailment in the wind farm, the optimal speed control takes the maximization of the rotor's rotational kinetic energy as the goal to determine the active power regulation commands for each WT, thereby reducing the wind energy loss. Camblong et al. [178] developed and analyzed a linear quadratic gaussian controller, and they also used pitch angle and generator torque as control parameters to control a WT electrical power and rotational speed. The electrical power and rotational speed references are produced at higher control levels based

on frequency changes and wind speed, thus reducing the drive fatigue load of the source side and optimizing the grid primary frequency control of the grid.

## 5. Discussion and Conclusions

The excitation and mutual coupling from the wind source and grid make the WT's load characteristics more complex concerning the large-scale development of wind farms in complex terrain and the improvement of wind–grid connection requirements. An accurate understanding of the load characteristics of different parts of the WT and its transmission mechanism between various parts is of momentous scientific significance to improve the design and the practical operation reliability of various components of the WT. This paper reviews the research status of WT's load characteristics, motivated by the variable parameters from both the source side and grid under complex terrain. For this purpose, this paper summarizes the research achievements of WT load characteristics under dual source–grid variable excitation on complex terrain. Firstly, according to the source side, grid side, and transfer coupling, the different research models of diverse load are summarized, and the characteristics and transfer mechanism of diverse load are analyzed. Moreover, the variation characteristics of fatigue load and aerodynamic load are analyzed, and the different research models of the load are summarized. Besides, the existing problems were pointed out, and the advantages/disadvantages of the existing improvement schemes are analyzed. Finally, the model of the transmission chain and control strategy are described under multi-factor coupling. This review provides a theoretical foundation and indicates directions for the safety design and reliable operation research of large WTs on complex terrain, with crucial scientific significance.

The main conclusions are summarized as follows:

1.  WT loads are varied in form and complex in source, and they can be divided into different situations according to different classification standards. The classification of load in this paper mainly has the following basis: the time-varying characteristics of load, the design of WT parts, source of load, and property of load.
2.  According to the source side, grid side and transfer coupling, the different research models of diverse load were summarized, and the characteristics and transfer mechanism of diverse load are analyzed.
3.  Studies on the WT's load characters considering single factors are abundant. However, a WT operating on complex terrain is affected by dual source–grid variable excitation and transference, which aggravated the complexities of the load. Meanwhile, most of the studies are based on simulation or wind tunnel tests, which are not realistic to solve the actual layout of wind farms through these studies. When optimizing the positions of WTs in complex terrain, all optimization simulations require numerous times of calculation for the wake flow. With the development of wind farms in complex terrain and the increasing flexibility of WTs, the research on load characteristics and transfer mechanisms of WTs in heterogeneous flow field needs to be further studied.
4.  The dynamic characteristics of WT will affect the grid-connected quality of wind power, and the interference and faults of the power grid will also affect the mechanical and electrical components and mechanical components of wind power. The changes of grid behavior are mainly reflected in the voltage, the power flow, and the system frequency. In the transition process of power grid voltage sag and recovery, the electromagnetic torque of the motor will fluctuate greatly, which will inevitably bring about the oscillation of the torque after the fault process and fault removal, and it may further impact the mechanical components, such as gear case, affecting the operation and life of WT. However, at the same time, it may influence the stability of generator output power and speed.
5.  The commonly used model of the transmission chain with the multi-mass block equivalent model and the FMB model was reviewed. However, the location of wind field and the design of WT often pay more attention to the source side. To increase

the power generation and reduce the load under multi-factor coupling, one needs appropriate control strategies.

6.  A new research idea of 'comprehensively considering the coupling effects of source and network factors, revealing WT load characteristics and transmission mechanism' is summarized.

The above discussion and conclusions collated and analyzed the WT's load characteristics, excited by the wind and grid in complex terrain. These provide researchers in related fields with coupled source–grid bipartite information, and that has an important scientific role, which contributes to the development of this topic. Further, it improves the reliability of WT design and operation, and, moreover, it promotes the efficient and safe application of wind energy.

**Author Contributions:** Acquisition of data and data curation, W.L. (Wei Li); Writing-Original draft, data analysis and Methodology, S.X.; Investigation, B.Q.; Paper idea provider and supervisor of the experiments and paper organization, Funding acquisition, Supervision and visualization, X.G.; Date curation, X.Z.; Formal analysis, Z.S.; Funding acquisition, W.L. (Wei Liu); Methodology and investigation, Q.H. All authors have read and agreed to the published version of the manuscript.

**Funding:** This research was funded by the National Natural Science Foundation of China (No. 52076081), the Fundamental Research Funds for the Central Universities (No. 2020MS107), the Research Institute for Sustainable Urban Development (RISUD) with account number of BBW8 of The Hong Kong Polytechnic University, the development of customized wind power prediction system under terrain wake coupling (No. SGXJ XNOOTSJS2200142), (No. SGXJ XN00TSJS2200143) and The APC was funded by Xiaoxia Gao.

**Institutional Review Board Statement:** Not applicable.

**Informed Consent Statement:** Not applicable.

**Conflicts of Interest:** The authors declare no conflict of interest.

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
