# Peer review of "Large-Scale Wind Turbine’s Load Characteristics Excited by the Wind and Grid in Complex Terrain: A Review"

_sustainability, doi:10.3390/su142417051_

Round 1

Reviewer 1 Report

Dear authors,

The article in it is actual form well organised,  and present valuable information. It requires English check on typos and spell check.

Congratulations! 

Author Response

Response : Thank you very much for your valuable suggestions. The spelling of words in the article has been checked word by word.

Reviewer 2 Report

The topic is interesting and very much relevant to the current context, moreover, the review paper has wider theoretical and practical applications. The authors have put their best efforts to execute this paper. However, I have the following reservations and suggestions for the sake of improvement of the undertaken study:

1) The logical sequence of the abstract should be as 1) objectives, 2) methodology, 3) Findings, 4) conclusion and 5) implications. Thus, the authors should also rewrite the abstract in this sequence. The authors should mention the review analysis method in the methodology, after findings conclusion and then please describe important implications. 

2) The authors did not establish the motivation, significance, and novelty of the undertaken study. The authors are suggested to improve this important factor in the "Introduction" section. The background of research should also be presented in the section. The structure of the review paper should also be presented in the end of Introduction section.

3) The literature should be presented in a proper heading, and should be separately from the introduction, and it should be presented in an audit form, and should be linked with the objectives of the current paper.

4) The material methods OR methodology section should be added in a proper heading, and it should also contain on selection of papers, flow diagram and method of review of the papers etc. 

5) The discussions section should be added before conclusions; the discussions section provides the opportunity for the authors to sell their idea to the readers. The discussions be should be complemented with the previous literature.

6) The conclusion should be added after the discussions section, conclusion is always one step ahead of findings. 

7) The practical, theoretical and societal implications should be discussed after the conclusion, and in the light of the conclusion and discussions. 

8) Minor spelling and grammatical mistakes should be improved.

Author Response

Point 1: The logical sequence of the abstract should be as 1) objectives, 2) methodology, 3) Findings, 4) conclusion and 5) implications. Thus, the authors should also rewrite the abstract in this sequence. The authors should mention the review analysis method in the methodology, after findings conclusion and then please describe important implications. 

Response 1: We appreciate your valuable suggestions. Rewrote the abstract in a logical order. In particular, the methodology of the review and significance at the end was added.

Point 2: The authors did not establish the motivation, significance, and novelty of the undertaken study. The authors are suggested to improve this important factor in the "Introduction" section. The background of research should also be presented in the section. The structure of the review paper should also be presented in the end of Introduction section.

Response 2: Your suggestions are precious and instructive. An innovative statement has been added to the penultimate paragraph of the Introduction to highlight the motivation, significance and novelty of the research conducted. The background of the study and the structure of the article in this section also have been optimized appropriately .

Point 3: The literature should be presented in a proper heading, and should be separately from the introduction, and it should be presented in an audit form, and should be linked with the objectives of the current paper.

Response 3: Thanks for your professional review comments. The introduction is separated into 3 parts: Research background, literature review and innovation points and paper structure. Meanwhile, the part of literature review elaborates in an audit manner from two aspects of the source side and grid side linked with the objectives.

Point 4: The material methods OR methodology section should be added in a proper heading, and it should also contain on selection of papers, flow diagram and method of review of the papers etc. 

Response 4: Appreciate your suggestions. Inappropriate headings have been properly revised as required.

Point 5: The discussions section should be added before conclusions; the discussions section provides the opportunity for the authors to sell their idea to the readers. The discussions be should be complemented with the previous literature.

Response 5: Your points are instructive and important in terms of how to facilitate the reader's reading. Discussion of " Large-scale Wind Turbine’s Load Characteristics Excited by the Wind and Grid in Complex Terrain " precedes the conclusion of the article, and "Discussion" has been added to the last title.

Point 6: The conclusion should be added after the discussions section, conclusion is always one step ahead of findings. 

Response 6: We appreciate your valuable suggestions.The discussion section has been placed before the conclusion section.

Point 7: The practical, theoretical and societal implications should be discussed after the conclusion, and in the light of the conclusion and discussions. 

Response 7: We appreciate your review comments. A new paragraph has been added after the conclusion to discuss the practical, theoretical and social impact of the topic

Point 8: Dear authors, The article in it is actual form well organized, and present valuable information. It requires English check on typos and spell check.

Response 8: The spelling of words in the article has been checked word by word. Thank you very much for your valuable suggestions!
